# Focus on Anti-Tumour Necrosis Factor (TNF)-α-Related Autoimmune Diseases

**DOI:** 10.3390/ijms24098187

**Published:** 2023-05-03

**Authors:** Loris Riccardo Lopetuso, Claudia Cuomo, Irene Mignini, Antonio Gasbarrini, Alfredo Papa

**Affiliations:** 1Center for Diagnosis and Treatment of Digestive Diseases, CEMAD, Gastroenterology Department, Fondazione Policlinico Gemelli, IRCCS, 00168 Rome, Italy; lopetusoloris@libero.it (L.R.L.); irene.mignini@gmail.com (I.M.); antonio.gasbarrini@unicatt.it (A.G.); 2Department of Medicine and Ageing Sciences, “G. d’Annunzio” University of Chieti-Pescara, 66100 Chieti, Italy; 3Center for Advanced Studies and Technology (CAST), “G. d’Annunzio” University of Chieti-Pescara, 66100 Chieti, Italy; 4Department of Translational Medicine and Surgery, School of Medicine, Catholic University, 00168 Rome, Italy

**Keywords:** infliximab, adalimumab, etanercept, certolizumab pegol, golimumab, autoimmune diseases, inflammatory bowel disease, paradoxical reactions

## Abstract

Anti-tumour necrosis factor (TNF)-α agents have been increasingly used to treat patients affected by inflammatory bowel disease and dermatological and rheumatologic inflammatory disorders. However, the widening use of biologics is related to a new class of adverse events called paradoxical reactions. Its pathogenesis remains unclear, but it is suggested that cytokine remodulation in predisposed individuals can lead to the inflammatory process. Here, we dissect the clinical aspects and overall outcomes of autoimmune diseases caused by anti-TNF-α therapies.

## 1. Introduction

Anti-tumour necrosis factor (TNF)-α agents are increasingly being used for the treatment of patients affected by inflammatory bowel disease (IBD) such as ulcerative colitis (UC) and Crohn’s disease (CD), as well as rheumatologic and dermatological disorders such as rheumatoid arthritis (RA), juvenile idiopathic arthritis (JIA), ankylosing spondylitis (AS), psoriatic arthritis (PsA), and psoriasis. Notably, the majority of these drugs can neutralise both forms of TNF-α, either by binding the trans-membrane form (m-TNF-α) or by blocking the soluble form (s-TNF-α) [1]. Only infliximab (IFX) and adalimumab (ADA) also induce apoptosis in T cells and monocytes [1,2]. Conversely, etanercept (ETA) is a recombinant, covalently bound dimer of the soluble p75 TNF receptor fused to the Fc portion IgG1, which binds to the soluble TNF-α form, but not the membrane-bound TNF-α [3]. Among the TNF-α blockers, only ETA is not approved for treating IBD patients, both CD and UC. In contrast, the last authorised agent, golimumab (GOL), is given only for UC but not for CD, other than for other clinical conditions such as RA, PsA, AS, and polyarticular JIA. On the other hand, the fifth anti-TNF-α agent, certolizumab pegol (CER), is only approved for CD by the United States Food and Drug Administration (FDA) but not by the European Medicines Agency (EMA), though it is used for PsA, RA, SpA, and psoriasis. It is worth noting that ETA, GOL, CER, and ADA are given subcutaneously, while IFX is administered intravenously. However, it is worth mentioning that subcutaneous IFX has recently become available, but there is no data yet on possible paradoxical autoimmune manifestations involving it.

With the widespread use of anti-TNF-α agents, concerns about any unexpected effects have been raised. Indeed, over the past 20 years, the widening use of biologics in several autoimmune diseases has been related to a new class of adverse events called paradoxical reactions. These immune-mediated processes appear paradoxically during treatment and can virtually affect several organs, including the skin, liver, lungs, kidneys, peripheral and central nervous system, vascular system, and bowels (Table 1). Despite the significant role of these drugs in suppressing TNF-α, a crucial cytokine in the development of the inflammatory process, a growing number of reports of paradoxical autoimmune processes associated with anti-TNF-α agents have emerged. This review aims to dissect the clinical aspects and overall outcomes of autoimmune diseases caused by biological therapies.

## 2. Dermatological Manifestations

Anti-TNF-α therapy can cause a wide range of dermatological conditions, including local skin irritation or reaction, increased skin infection rates, psoriasis, eczema, anti-TNF-induced cutaneous lupus erythematosus (ATIL), and alopecia areata (AA). Other reactions rarely occur, such as granuloma annulare, lichen planus, vitiligo, and cutaneous vasculitis. In patients with IBD, some of the above complications, such as erythema nodosum and pyoderma gangrenosum, could be extraintestinal manifestations related or not to clinical exacerbations. Thus, temporal associations with biological therapy could differentiate between disease- and drug-related complications. These reactions are managed by stopping anti-TNF-α treatment, switching to a different TNF-α inhibitor or ustekinumab, switching to another systemic immunosuppressant for treating the underlying dermatologic condition, or continuing anti-TNF-α therapy with the addition of topical or systemic skin-directed treatments, depending on the severity and extension of the cutaneous lesions.

### 2.1. Psoriasis

Psoriasis is a chronic autoimmune disease characterised by raised red, itchy, scaly patches, most commonly on the knees, elbows, trunk, and scalp. TNF-α antagonist-induced psoriasis, which can emerge de novo or exacerbate a pre-existing form, is the most common dermatological adverse reaction linked to anti-TNF therapy [4,5]. In most cases (80%), these adverse events represent a new disease, while others exacerbate a pre-existing illness. In a recent case series involving 150 patients with psoriasis who received treatment with TNF-α inhibitors [6], 10 patients developed paradoxical psoriasis (PP) without stopping anti-TNF-α therapy. Nonetheless, 80% of them achieved remission again after the aggravation of symptoms without any other serious adverse events. Several studies have been conducted to identify the main characteristics and pathogenesis of these specific lesions, concluding that the pathogenesis of PP could be different from that of classical psoriasis. In fact, it has been shown that classical psoriasis is a T-cell-mediated autoimmune disease driven by TNF.

In contrast, PP is caused by the absence of TNF-α and represents type I interferon (IFN)-driven innate inflammation [7]. Stoffel sought to characterise anti-TNF-induced inflammatory skin lesions on a histopathologic, cellular, and molecular level compared to psoriasis, eczema (atopic dermatitis), and healthy control skin [8]. Histopathologic evaluation, gene expression, and computer-assisted immunohistologic studies have been performed on 19 skin biopsies from IBD (*n* = 17) and rheumatoid arthritis (*n* = 2) patients with new-onset inflammatory skin lesions during anti-TNF-α therapy [8], evidencing the new onset of psoriasiform, eczematous, and scalp anti-TNF-induced skin lesions. While the histopathologic aspects were similar, these reactions were considered distinct immunological entities with Th1/IFN-γ/IFN-α characteristics; all the anti-TNF-induced lesions showed a more robust IFN-γ activation than in psoriasis or eczema [8]. This signature was consistent with an increased concentration of Th1 but not Tc1 or natural killer (NK) cells. Another study investigated the immunological and genetic profiles of three hidradenitis suppurativa (HS) patients without psoriasis who developed paradoxical psoriasiform reactions following anti-TNF-α therapy with ADA. These skin manifestations showed immunological features common to the early phases of psoriasis [9]. Among them, LT-α and LT-β, as well as IFN-κ and IFN-λ1, were identified as new innate mediators potentially involved in the induction process, mainly characterised by innate cellular and molecular players [9].

IFX and ADA are the most frequently associated with psoriasis among anti-TNF drugs. Collamer et al. reviewed 207 cases of new-onset psoriasis, of which 43% of patients had RA, 26% seronegative SpA, and 20% IBD [10]. The most frequently implicated drug was IFX (50% of cases), followed by ADA and ETA, while reactions to Certolizumab or Golimumab were infrequent [10]. In this scenario, in IBD, the onset of PP has been associated with the female sex, comorbidities, and the use of ADA [11]. In addition, a high incidence of anti-TNF-associated psoriasis in patients treated with IFX or ADA was confirmed in further analyses, which also documented an association with female gender, foregut disease location, fistulising, and stricturing disease behaviour [12,13].

Besides the adult setting, little is known about paediatric IBD patients. PP appears to be the most frequent side effect in paediatric CD treated with IFX. Perianal lesions and young age at anti-TNF initiation represent the most significant risk factors [14].

### 2.2. Anti-TNF-Induced Lupus (ATIL)

Following the introduction of anti-TNF, ATIL has become a new and increasingly recognised clinical entity [15]. All anti-TNF-α agents have been associated with an increased risk of autoantibodies. Up to 49% of patients on these medications develop raised antinuclear antibodies (ANAs) or anti-double strain (ds) DNA. However, despite the high antibody prevalence, clinical manifestations of ATIL remain rare [16].

Overall, the first case of drug-induced lupus (DILE) was reported in 1945 [17] as a complication of sulfasalazine therapy, described as a milder disease form than classical systemic lupus erythematosus (SLE), which resolved spontaneously after stopping treatment. DILE is an autoimmune disorder that can present with joint pain, skin involvement, fatigue, and serositis. Conversely, ATIL appears distinct from DILE, with a phenotype similar to idiopathic SLE. The incidence/prevalence of dsDNA antibodies and hypocomplementemia is greater in ATIL, while anti-histone antibodies, the serological hallmark of classical DILE, are less commonly found. The most common symptoms of ATIL are arthralgias and arthritis, haematological abnormalities, and a skin involvement similar to SLE [18]. Cerebral and renal involvement has been reported more frequently in ATIL than in classical DILE. In this scenario, the central nervous system (CNS) is not often involved, and neuropsychiatric symptoms such as depression and suicidal ideations are not commonly seen in ATIL patients [19].

A large Australian cohort study including 454 patients treated with anti-TNF-α therapy (300 with IFX and 154 with ADA) reported an incidence of ATIL of 5.7% for IFX and 0.6% for ADA [20]. These rates appear much higher than those in post-marketing studies. The diagnostic criteria for ATIL diagnosis are a temporal relationship between symptoms and anti-TNF-α therapy administration and resolution of symptoms following cessation of the offending medication; at least one serologic American College of Rheumatology criterion of SLE; at least one non-serological criterion such as arthritis, serositis, or a rash. Another dedicated study assessed the clinical characteristics of three IBD patients treated with IFX [21]. The first patient had a classical DILE with thrombocytopenia that resolved after IFX discontinuation, the second case experienced symmetric polyarthritis of 14 joints in RA-like distribution accompanied by lymphopenia, and the third one had severe serositis with ascites, pleural and pericardial effusions, and pancytopenia [21]. In this patient, ATIL coexisted with anti-TNF-induced hepatitis [21]. The occurrence of both lupus-like syndrome and hepatitis following anti-TNF-α therapy in the same patient is infrequent. According to these reports, the clinical presentations of ATIL can vary from mild cases to severe adverse events. However, specific diagnostic criteria for ATIL have not been recognised yet. Although the primary evidence of lupus development comes from studies involving IBD patients, this feature is not exclusive since it can also occur in other chronic inflammatory conditions [22]. Of note, while stopping anti-TNF-α therapy has. been shown to be beneficial, reintroducing another anti-TNF-α does not cause significant consequences [23]. Only a few severe cases characterised by systemic involvement have required conventional therapy for SLE to achieve complete remission.

### 2.3. Other Skin Manifestations

Alopecia areata (AA) can be described as a non-scarring inflammatory pathology with a sudden onset and rapid hair loss in some regions of the scalp and body. It can be divided into alopecia focalis (monolocularis or multilocularis), which can occur in one or multiple areas of the scalp; alopecia totalis, which happens all over the scalp; alopecia universalis, which is a complete loss of scalp and body hair. It is linked to a multifactorial process involving an immuno-mediated response and the overexpression of proinflammatory cytokines such as TNF-α, interleukin-1-α, and IFN-γ. According to this evidence, a role for anti-TNF in the pathogenesis of AA seems unlikely. Nevertheless, few reports have shown this connection [24], such as the case of a 50-year-old man with psoriatic arthritis who exhibited alopecia universalis with concomitant onychodystrophy three months after ADA initiation [25]. In this setting, anti-TNF therapy could have caused cytokine dysregulation that could trigger these manifestations.

Among dermatological adverse effects, reports have associated TNF-α inhibitors with the onset of lichenoid eruption (LE). Lichen planus is a recurrent, itchy inflammatory rash with single, small, polygonal papules that can merge to form rough, scaly plaques, often accompanied by injury to the oral cavity or genitals. Diagnosis is usually clinical and can be supported by a skin biopsy. Treatment generally requires topical or intralesional corticosteroids. A case report described three IBD patients who developed oral lichen planus after starting anti-TNF therapy [26]. Another paper documented five patients with LE following anti-TNF-α treatment for ankylosing spondylitis [27]. In these cases, the development of an inflammatory reaction is somewhat surprising since TNF-α is involved in the pathogenesis of lichen planus, and TNF-α inhibitors are efficacious when used to treat refractory cases. The potential cause could be linked to the anti-TNF-mediated remodulation of cytokine release with subsequent increasing levels of type I IFN and the activation of T and dendritic cells [27].

Vitiligo is a long-term skin condition characterised by patches of the skin losing its pigment. Hypotheses for its pathogenesis include immune-mediated destruction of epidermal melanocytes [28]. Vitiligo treatment involves general, non-targeted immunosuppressants that provide only modest efficacy. TNF-α has been shown to contribute to depigmentation. Indeed, TNF-α levels in vitiligo lesions are higher than those in non-lesional skin and are closely related to disease activity [29,30]. Following this rationale, anti-TNF treatments have been described for active vitiligo.

Conversely, new-onset cases following biological treatment have emerged [31]. In this setting, the risk of developing vitiligo and AA in patients with AS, CD, or UC treated with anti-TNF therapy compared to those without anti-TNF agents has been evaluated using the Korea National Health Insurance Claims databases [32]. A significantly increased risk of vitiligo was observed in the anti-TNF-α group compared to the unexposed group. In subgroup analyses, younger patients and those treated with ETA showed the highest risk. No statistically significant difference was observed in the development of AA between the two groups. Accordingly, TNF inhibition may shift cytokine patterns with the subsequent recruitment of autoreactive T cells to the epidermis and the destruction of melanocytes.

In addition, several cases of anti-TNF-mediated dermatomyositis/polymyositis have been described [33]. Here, the pathophysiological mechanism is unknown, but increasing interferon-gamma levels following therapy could represent a crucial risk factor [34].

## 3. Liver Diseases

The primary concerns about anti-TNF-α therapy-related liver injury refer to the increased susceptibility towards chronic infections, especially towards the reactivation of the hepatitis B virus (HBV) [35]. Autoimmune hepatic toxicity is less common, but several cases have been reported during post-marketing surveillance. Although not frequent, autoimmune disorders are nonetheless significant [35,36]. The drug most often involved is IFX and, to a lesser extent, ADA and ETA, with a solid female predominance [36,37]. The predominant form of liver injury mediation is characterised by a hepatocellular profile with both serological and histological features of autoimmunity. In most reported cases, biopsy shows chronic lymphoplasmocytic infiltrate and interface hepatitis with altered liver function tests associated with positive autoantibodies, such as ANA, anti-smooth muscle antibodies (ASMAs) or anti-dsDNA, or increased immunoglobulins [38]. These findings support the autoimmune origin of anti-TNF-related hepatitis, which resembles the autoimmune hepatitis type I or hepatitis in the context of a lupus-like syndrome. Moreover, anti-TNF-induced liver diseases include hepatocellular injury without autoimmunity and cholangiopathy [39,40].

Clinical manifestations range from asymptomatic abnormalities in liver function tests, self-limiting after drug discontinuation, to more severe damage, requiring corticosteroid therapy. Tobon et al. described two cases of severe liver disease: a 39-year-old woman who was treated with IFX for eight months and developed cholestatic liver disease and hepatic failure with hepatic encephalopathy and ascites and finally underwent liver transplantation, and a 54-year-old woman who was diagnosed with acute hepatitis with positive ANAs after 17 months of IFX therapy and was treated with prednisolone and azathioprine [41]. Cases of fulminant hepatitis have also been reported, though rarely, leading to transplantation or death. In clinical practice, differentiating between autoimmune hepatitis and drug-induced hepatitis is challenging, with often indistinguishable symptoms, serology, and histology. Usually, patients with anti-TNF-induced liver injury do not relapse after the resolution, with or without immunosuppressive therapy, and ANAs disappear after steroid therapy [42,43]. Here, the pathogenesis is still unclear. However, it has been hypothesised to be more complex than just a class effect since some patients with IFX-mediated autoimmune hepatitis well tolerated other TNF-α inhibitors [35,36,44,45]. It has been suggested that the absence of cross-toxicity between anti-TNF-α drugs and differences in the clinical response may be due to their different molecular structure. This hypothesis could also explain why IFX, the only chimeric mouse–human antibody, is the anti-TNF-α most prone to causing autoimmune liver injury [46]. Thus, a pathogenic role could be played by specific antibodies against IFX. Moreover, IFX could impair the physiological suppression of auto-reactive B cells, leading to an increased lymphocyte presence. It may provoke cellular apoptosis and elicit self-antigens release, causing the loss of tolerance towards the patient’s hepatocytes [35,36,47,48].

## 4. Interstitial Lung Diseases

Anti-TNF-related lung autoimmune disorders are mainly represented by interstitial lung diseases (ILDs) and pulmonary fibrosis, the most severe manifestation. However, the association between these drugs and developing or exacerbating ILDs is still controversial [49]. Experimental evidence shows that TNF-α antagonists may improve or paradoxically induce ILDs [49,50]. ILDs include heterogeneous lung disorders with essential and potentially severe manifestations of chronic inflammation, ranging from asymptomatic to potentially life-threatening. Even if an ideal treatment is not well established, immunosuppressive drugs and TNF-α blockers are administrated to reduce inflammation and stabilise the progression of ILDs [51].

In contrast, case reports have described ILDs and pulmonary fibrosis as paradoxical side effects of anti-TNF-α agents. These drugs may both induce new lung injury and worsen pre-existing ILDs. Most available data refer to patients affected by rheumatoid arthritis [52,53]. The drugs most frequently involved are IFX and ETA, followed by ADA. Interstitial pneumonia appears to be the most common pulmonary autoimmune complication of these drugs [54,55,56]. Interestingly, Komiya et al. reported the case of a man with a history of RA and lung involvement. His pulmonary lesions and arthritic symptoms first improved with ADA, but then he developed interstitial pneumonia, showing the simultaneous conflicting actions of anti-TNF agents [57].

Similarly, even if less frequently, developing or exacerbating ILDs under biological therapy has also been reported in diseases other than rheumatoid arthritis, such as psoriatic arthritis/psoriasis, ankylosing spondylitis, systemic sclerosis, and IBD [55,58,59]. The case of a 53-year-old man with UC was described in this setting. He reported a fatal acute interstitial pneumonitis after completing an accelerated induction course of IFX. Four similar cases of IFX-induced interstitial lung disease in UC patients were also evidenced, but they had a successful outcome after IFX discontinuation [60]. However, a large cohort study conducted on 8471 patients with RA, ankylosing spondylitis, psoriatic arthritis, psoriasis, and IBD showed that anti-TNF-α therapy does not increase the occurrence of ILDs among patients with autoimmune diseases when compared to non-biologic treatments [61].

Conversely, it has been suggested that anti-TNF-α might exacerbate pre-existing ILDs compared to non-anti-TNF-α biological therapy [62]. In a cohort of 40 patients, most cases showed that anti-TNF-α was harmful to patients with ILD, with a 35% mortality rate. Age, use of antibiotics, and association with azathioprine were considered risk factors. In addition, adverse ILD events tended to appear in female patients with a long history of RA. The most common clinical manifestations in ILD adverse events were shortness of breath and sudden dyspnoea, followed by non-productive cough [63]. Radiologically, the non-specific interstitial pneumonia pattern was the most common imaging type [63]. Translational studies have demonstrated that despite a pathogenic role of TNF-α in favouring chronic lymphocytic alveolitis [64] and fibroblast activation [65], and vice versa, a lack of TNF-α seems to prevent bleomycin-induced lung fibrosis [66]. Further studies are needed to clarify the uncertain role of TNF-α inhibitors in lung disorders and help treatment decisions. In particular, more caution should be taken when prescribing anti-TNF to RA patients, especially older patients with a more extended RA history or RA-associated pulmonary disease.

## 5. Renal Disorders

Compared to other anti-TNF-mediated adverse events, autoimmune renal disorders are uncommon. They are most often described in patients affected by rheumatic diseases, especially rheumatoid arthritis, ankylosing spondylitis, and psoriatic arthritis, but also in other conditions, such as psoriasis and IBD. ETA is the drug most frequently implicated, followed by ADA and IFX [67]. Renal alterations can appear as isolated manifestations or as part of more complex disorders such as systemic vasculitis or SLE. Different types of vasculitis with renal involvement have been described, notably ANCA-associated vasculitis, both p-ANCA and c-ANCA, which are often characterised by a histopathological pattern of necrotising crescentic glomerulonephritis [68,69], and Henoch–Schönlein purpura, which are often related to mesangial glomerulonephritis [70]. Biopsies performed in SLE-associated glomerulonephritis have shown class IV or, less frequently, class III lupus glomerulonephritis. These patients had increased levels of antinuclear antibodies (ANAs), and some also had increased anti-dsDNA antibodies [71]. Isolated renal disorders show various manifestations, including membranous, mesangial, necrotising crescentic, minimal changes in glomerulonephritis, and focal segmental glomerulosclerosis [72].

Acute non-granulomatous tubulointerstitial nephritis has also been reported in an HLA-B27-positive patient with axial spondylarthritis and CD treated with ADA [73]. Sporadic cases of renal sarcoidosis have also been reported [74,75].

## 6. Systemic and Cutaneous Vasculitis

An increasing number of therapeutic agents have been associated with vasculitic syndromes. Among these, TNF-α inhibitors play a central role. Cutaneous manifestations with palpable purpura and leukocytoclastic vasculitis (LCV) are usually the most common upon histological examination, while systemic vasculitis mainly affects the peripheral nervous system and the kidneys [76,77,78]. Other sites of vasculitic involvement can include the lungs, central nervous system, gallbladder, and coronary arteries. LCV is an uncommon small-vessel vasculitis known as hypersensitivity vasculitis and hypersensitivity angiitis. It is an inflammatory process that can affect the small blood vessels of any organ. When acute, cutaneous leukocytoclastic vasculitis is the most frequent feature. Potential LCV causes include autoimmune diseases (i.e., RA, SLE, Sjogren, and Henoch–Schönlein purpura), infection, and drug-mediated immune reactions, especially TNF-α inhibitors. Some authors have suggested that host antibodies developed against anti-TNF-α agents may cause an immune-complex-mediated hypersensitivity reaction [79]. In most cases, drug discontinuation and sometimes corticosteroid treatment lead to the complete resolution of the cutaneous manifestations. In this setting, individual genetic susceptibility could explain the different disease localisations between cutaneous and systemic vasculitis.

## 7. Neuropathy

Demyelination has been described as a consequence of anti-TNF-α therapy [80]. A review of the FDA Adverse Event Reporting System showed 772 reports of neurological complications, especially in patients with RA (50.9%) and IBD (18%). ETA, followed by IFX, resulted in the most involved drugs [81]. Peripheral neuropathy was the most frequent manifestation, followed by CNS and spinal cord demyelination [81,82]. However, CNS demyelination after anti-TNF-α treatment is not necessarily associated with the duration of therapy and drug discontinuation does not always coincide with the resolution of symptoms.

## 8. Sarcoidosis

Sarcoidosis is a systemic inflammatory disorder characterised by non-caseating granuloma, which commonly affects the lungs, lymph nodes, and salivary glands, but rarely other organs (i.e., liver, spleen, eyes, heart, and skin). In recent years, TNF-α inhibitors have been used for its treatment. However, there have been isolated cases of sarcoidosis occurrence during anti-TNF therapy. Daïen et al. estimated a frequency of 1:2800 [83]. In most cases, sarcoidosis occurs after treatment with ETA [83], but it has also been described with ADA and IFX [84,85,86]. The underlying disease is often RA, followed by AS and PsA. Symptom improvement is registered in most patients after drug exclusion, with only a few cases requiring steroid therapy [84,85,86,87,88].

Interestingly, Isshiki et al. described the case of an RA patient who developed sarcoidosis six months after the introduction of ETA and improved without therapeutic intervention after its cessation [89]. In this case, *Propionibacterium acnes* was detected in the granulomas, suggesting that *P. acnes* infection may show aberrant responses to drugs such as TNF-α inhibitors and immune checkpoint inhibitors, consequently leading to drug-induced sarcoidosis [89]. Further potential hypotheses on the pathogenesis of anti-TNF-mediated sarcoidosis have been proposed. It has been shown that the mechanisms of action of ETA and IFX can produce different TNF-α concentrations in the tissue, and higher levels of biologically active TNF-α from ETA therapy may foster the formation of granulomas [79]. This phenomenon may be due to the molecular structure of ETA. While monoclonal antibodies (i.e., IFX and ADA) can neutralise both soluble and membrane forms of TNF-α, ETA neutralises just the soluble form. Only “partial” inhibition can lead to a redistribution of TNF-α in areas of elevated concentration, such as the lungs and lymph nodes [90,91].

Furthermore, several studies have suggested that monoclonal antibodies, in contrast to ETA, inhibit the expression of the IL-17 and IFN-γ cytokines intensely involved in granuloma formation. In contrast, some reports have shown sarcoid-like reactions in patients treated with IFX and ADA [92,93]. In these cases, cytokine imbalance due to long-term TNF-α suppression could lead to those paradoxical reactions. Moreover, as shown for paradoxical skin reactions, anti-TNF-α can increase INF production, which is crucial for granuloma formation [94].

## 9. New-Onset IBD

Both paradoxical worsening of IBD and new-onset cases induced by anti-TNF-α agents have been reported in patients with different inflammatory diseases, especially AS, JIA, and psoriasis. CD has been observed more frequently than UC, with most cases attributed to ETA [95,96]. In contrast with the monoclonal antibodies, ETA is ineffective in treating IBD because of its inability to induce T cell apoptosis in the intestinal mucosa [97]. Some authors have suggested that paradoxical IBD could represent a subclinical condition unmasked by TNF-α inhibitors, particularly by ETA and less frequently by IFX [96,98,99,100]. The reason behind this different risk rate is again linked to their divergent mechanisms of action. IFX binds to both soluble and membrane TNF-α, while ETA binds to soluble TNF-α and lymphotoxin-α, causing a partial TNF-α neutralisation. IFX but not ETA can induce circulating and lamina propria T lymphocyte apoptosis by binding directly to transmembrane TNF-α and caspase-3 activation. Conversely, ETA can increase T lymphocyte levels in peripheral blood and the number of proinflammatory cytokines, such as IFN-γ, which can trigger intestinal inflammation. Despite the well-known concept that extraintestinal inflammatory diseases (i.e., AS, JIA, psoriasis, and PsA) can represent independent risk factors for the development of IBD, the fact that, after the interruption of ETA and the switch to a different anti-TNF-α or the swap to another biological agent, patients usually obtain an immediate improvement of gastrointestinal inflammation suggests the theory of a paradoxical effect [101,102].

## 10. Conclusions

Anti-TNF-α agents has significantly improved the outcomes of IBD and dermatological and rheumatological conditions. Paradoxical events due to these therapies are not commonly reported in the literature, especially compared to other side effects. In recent years, a growing number of cases and a more extensive range of clinical manifestations have been reported because of their increased use. The pathogenesis of the paradoxical reactions is not completely clarified yet. However, several mechanisms involved have been hypothesised and are summarised in Table 2. They include cytokine remodulation, which leads to the overexpression of some inflammatory molecules (i.e., IFN and IL-17/IL-23) able to trigger an inflammatory process, promoting the activation and amplification of pathogenic T cells, particularly in individuals with a genetic predisposition [10,103]. Besides this, most TNF-α inhibitors are immunogenic with consequent anti-drug antibody development and complex immune constitutions [104,105,106]. Anti-TNF-α agents can also impair the normal suppression of auto-reactive B cells, leading to an increased lymphocyte presence, or may provoke cellular apoptosis and the consequent release of self-antigens, leading to the loss of tolerance towards the patient’s cells [35]. Finally, the different molecular structures of the anti-TNF-α agents can lead to divergent effects. In this scenario, new studies should be conducted to extend our clinical knowledge on the pathogenic mechanisms underlying these paradoxical reactions to prevent them and to obtain an overall better management of IBD patients.

## Figures and Tables

**Table 1 ijms-24-08187-t001:** Autoimmune clinical manifestations caused by anti-TNF-α agents.

Autoimmune Clinical Manifestation	Most Involved Anti-TNF-α	Most Involved Disease/ Type of Patients	Treatment	Safe Switch to Alternative Anti-TNF-α Reported
Psoriasis	IFX	RA	Stop Anti-TNF-α	
Anti-TNF-induced Lupus	IFX	-	Stop Anti-TNF-α	
Vitiligo and AA	ETA	Young patients	Stop Anti-TNF	
Autoimmune Hepatitis	IFX	-	Stop Anti-TNF-α ± steroids	√
Interstitial Lung Diseases	IFX and ETA	RA	Stop Anti-TNF-α	
Renal Diseases	ETA	RA	Stop Anti-TNF	
Vasculitis	\	-	Stop Anti-TNF-α ± steroids	
Demyelination	ETA	RA	Stop Anti-TNF-α ± steroids	
Sarcoidosis	ETA	RA	Stop Anti-TNF-α ± steroids	
New-onset IBD	ETA	AS	Stop Anti-TNF-α	√

Abbreviations: TNF-α: tumour necrosis factor-α; IFX: infliximab; RA: rheumatoid arthritis; ETA: etanercept; AA: alopecia areata; AS: ankylosing spondylitis.

**Table 2 ijms-24-08187-t002:** Pathophysiology of paradoxical reactions and related clinical manifestations.

Primary Mechanism Involved in Paradoxical Reactions	Clinical Manifestation
Overexpression of IFN-γ and homing of Th1 cells	Psoriasis and other cutaneous manifestations (alopecia areata, lichenoid eruption, and vitiligo) and new-onset IBD
Activation of IL-23/Th-17 axis	Psoriasis and sarcoidosis
Anti-drug antibodies formation	Anti-TNF-induced lupus, systemic and cutaneous vasculitis, interstitial lung disease, and glomerulonephritis
Blockade of soluble TNF-α by ETA	Sarcoidosis, new-onset IBD, and demyelination
Decrease of TNFR2 receptors	Demyelination

## Data Availability

Not applicable.

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
