# Peer review of "Focus on Anti-Tumour Necrosis Factor (TNF)-α-Related Autoimmune Diseases"

_ijms, 2023, doi:10.3390/ijms24098187_

Round 1

Reviewer 1 Report

The author reviewed the most recently clinical research about anti-tumor necrosis factor (TNF)-α reagents, which is more and more widely used in chronic inflammatory diseases. The comprehensive adverse effects caused by anti-TNF-α reagents that the author listed are highly appreciated. But here are still several comments:

1.      The complexity of the adverse effects caused by different anti-TNF-α reagents and reports by multiple clinical researching works showed varied results in one aspect of side effects.

My comment is to create a table to show the readers more directly what is the most correlated side effects to a specific reagent. Please list the medicine name, treatment for what type of disease (might be administrated in different disease conditions), what type of side effect is most commonly observed in clinical, the potential underlying mechanism, the method to prevent/decrease the side effect, what type of people is most easily affected. Or, giving a cartoon picture to link the medicines mentioned in the author’s keywords portion to the side effects in treating what type of diseases.

2.      Please discuss more about this research work how to guide clinical medication of anti-TNF-α reagents, like the comorbidities, complications, age, sex, the time of medication (onset of the disease); and the further cares in using the anti-TNF-α reagents.

3.      Please reconsider the logic between these two paragraphs: the last paragraph on page 2 and the first graph on page 3, eg: “Finally” and “In contrast”.

Author Response

We thank the Reviewer for his helpful comments. We provided a point-to-point response.

  1. My comment is to create a table to show the readers more directly what is the most correlated side effects to a specific reagent. Please list the medicine name, treatment for what type of disease (might be administrated in different disease conditions), what type of side effect is most commonly observed in clinical, the potential underlying mechanism, the method to prevent/decrease the side effect, what type of people is most easily affected. Or, giving a cartoon picture to link the medicines mentioned in the author’s keywords portion to the side effects in treating what type of diseases.

Response: As requested by the Reviewer, we added a new Table (Table 1), including the type of autoimmune anti-TNF-α-induced clinical manifestation, the most frequently involved anti-TNF- agent, the most involved type of disease (or type of patients), the type of treatment of each autoimmune clinical manifestations and if a safety switch to alternative anti-TNF is reported in the literature.

  1. Please discuss more about this research work how to guide clinical medication of anti-TNF-α reagents, like the comorbidities, complications, age, sex, the time of medication (onset of the disease), and the further cares in using the anti-TNF-α

Response: We thank the Reviewer for his appropriate comment. As requested, we added some sentences to the introductory paragraph to clarify the differences in indications between the various TNF-α blockers available. Additionally, we have included information on the routes of administration of all available anti-TNF-α treatments. However, it is essential to note that there are no differences in the use of these agents about gender, disease onset, or comorbidities.

  1. Please reconsider the logic between these two paragraphs: the last paragraph on page 2 and the first graph on page 3, eg: “Finally” and “In contrast”.

     Response: We thank the Reviewer for his comment and changed the word “Finally” to “In fact”.

Reviewer 2 Report

The manuscript by Lopetuso et al. “Focus on Anti-Tumour Necrosis Factor (TNF)-α-Related Autoimmune Diseases” is a review focused mainly on the clinical aspects and outcomes of autoimmune diseases caused by anti-TNF-α therapies.

Although, as a literature review it has all the disadvantages of this type of article (i.e. random/subjective/unclear process of papers finding, covering of only several aspects of a topic), manuscript is well-written, presented in an intelligible fashion and the language is clear and correct. Most references are up to date and appropriate.

My only issue concerns Figure 1 which in my opinion is uninformative and redundant. Additionally, figure caption says “Organs involved in the paradoxical reactions caused by anti-TNF-α agents” however, on the figure we mainly see list of diseases. In my opinion these informations could be presented in the form of a table with headers such as for example “Organ; Disease;  Type of anti-TNF-α agents/drug used in the treatment; Paradoxical reactions caused by the treatment“.

Author Response

We thank the Reviewer for his helpful comments. We provided a point-to-point response.

  1. Although, as a literature review, it has all the disadvantages of this type of article (i.e. random/subjective/unclear process of papers finding, covering of only several aspects of a topic), manuscript is well-written, presented in an intelligible fashion and the language is clear and correct. Most references are up to date and appropriate.

Response: We thank the Reviewer for appreciating our work.

  1. My only issue concerns Figure 1 which in my opinion is uninformative and redundant. Additionally, figure caption says “Organs involved in the paradoxical reactions caused by anti-TNF-αagents” however, on the figure we mainly see list of diseases. In my opinion these informations could be presented in the form of a table with headers such as for example “Organ; Disease;  Type of anti-TNF-α agents/drug used in the treatment; Paradoxical reactions caused by the treatment“.

Response: We thank the Reviewer for his appropriate comment. As requested by the Reviewer, we have deleted Figure 1 and replaced it with a new Table (Table 1), including the type of autoimmune clinical manifestation induced by anti-TNF-α, the most frequently involved anti-TNF- agent, the type of disease most involved (or type of patients), the type of treatment of each autoimmune clinical manifestation and whether a safety switch to an alternative anti-TNF is reported in the literature.